# A Systematic Literature Review of Learning-Based Traffic Accident Prediction Models Based on Heterogeneous Sources

Pablo Marcillo * , Ángel Leonardo Valdivieso Caraguay and Myriam Hernández-Álvarez

Departamento de Informática y Ciencias de la Computación, Escuela Politécnica Nacional, Ladrón de Guevara E11-25 y Andalucía, Edificio de Sistemas, Quito 170525, Ecuador; angel.valdivieso@epn.edu.ec (Á.L.V.C.); myriam.hernandez@epn.edu.ec (M.H.-Á.)
* Correspondence: pablo.marcillo@epn.edu.ec

**Abstract:** Statistics affirm that almost half of deaths in traffic accidents were vulnerable road users, such as pedestrians, cyclists, and motorcyclists. Despite the efforts in technological infrastructure and traffic policies, the number of victims remains high and beyond expectation. Recent research establishes that determining the causes of traffic accidents is not an easy task because their occurrence depends on one or many factors. Traffic accidents can be caused by, for instance, mechanical problems, adverse weather conditions, mental and physical fatigue, negligence, potholes in the road, among others. At present, the use of learning-based prediction models as mechanisms to reduce the number of traffic accidents is a reality. In that way, the success of prediction models depends mainly on how data from different sources can be integrated and correlated. This study aims to report models, algorithms, data sources, attributes, data collection services, driving simulators, evaluation metrics, percentages of data for training/validation/testing, and others. We found that the performance of a prediction model depends mainly on the quality of its data and a proper data split configuration. The use of real data predominates over data generated by simulators. This work made it possible to determine that future research must point to developing traffic accident prediction models that use deep learning. It must also focus on exploring and using data sources, such as driver data and light conditions, and solve issues related to this type of solution, such as high dimensionality in data and information imbalance.

**Keywords:** machine learning; traffic accident prediction; heterogeneous sources; deep learning; neural network

## 1. Introduction

The World Health Organization (WHO), through the Global Status Report on Road Safety (GSRRS) 2018, affirms that the number of deaths by road traffic-related issues reached the number of 1.35 million people in 2016 [1]. Meanwhile, the Pan American Health Organization (PAHO) [2] affirms that traffic accidents were the second cause of death among young adults (15–29 years old) in 2016. However, the most concerning is that 47% of all people who died in traffic accidents are vulnerable road users, such as motorcyclists, cyclists, and pedestrians.

The implementation of technological infrastructure and the adoption of strict traffic policies have significantly reduced the accident rate. However, the number of victims is still high and beyond expectations. This situation partly happens because it is complex to determine the real causes of traffic accidents. In most cases, their occurrence depends on one or many of the following factors: mechanical problems, adverse weather conditions, mental and physical fatigue, negligence, potholes in the road, among others.

At present, the use of prediction models as mechanisms to mitigate mortality in traffic accidents is a reality. The results of these models are helping policymakers, transportation safety designers, and researchers to identify factors and make recommendations to make significant achievements in terms of the accident rate [3,4]. Some studies are being funded

by institutions or companies related to transportation, as in [4–9]. As soon as the prediction model can correlate information from heterogeneous sources, the model might infer accidents in a better way. However, this solution also brings along some issues to be resolved. For instance, some of them are the high dimensionality in data caused by information imbalance or the poor handling of large-scale datasets. In that way, the strategy to improve the prediction models must be focused on exploring other data sources to correlate them and finding strategies to resolve the issues related to this solution.

Since the models are generally fed with real data, the authors have resorted to government platforms and Internet services to collect data. The information from Internet services can be integrated into the prediction model to establish real-time information channels and improve their accuracy. However, this approach is not always feasible because the values and metrics of the different sources are not entirely comparable. In fact, there is much diversity in experimental design, acquisition protocol, equipment used, and data volume. For these reasons, it is important to highlight the current state of the development of learning-based traffic accident predictions and determine the main research challenges on this topic.

This paper presents a systematic literature review on learning-based traffic accident prediction models based on heterogeneous data sources. To elaborate on this review, we used the general guidelines proposed by Kitchenham's methodology [10,11]. The research questions and search strategy focused on identifying the most relevant features that influence the accuracy and performance of accident prediction models. With this analysis in place, our purpose is to respond to these concerns: How do human factors influence the occurrence of traffic accidents? How does the number of features used in a model affect its performance? How can information from different data sources be correlated? What are the solutions for the challenges that real-time prediction models face? What type of algorithms are best suited for traffic accident prediction models? Moreover, can the best model be determined using only the evaluation metrics? For this purpose, we study the different platforms, services, and simulators used to collect data related to traffic and driver behavior. Regarding the survey of traffic accident prediction models, our work includes a comparative study of models, selection algorithms, evaluation metrics, and the percentage of data used for training/validation/testing. Furthermore, the performance obtained by each model is registered, scored, and analyzed. Following this survey, we aim to find open challenges and research niches in the early prediction of traffic accidents to reduce the death of drivers and passengers.

This article is organized as follows: Section 2 presents the methodology used to elaborate this literature review, followed by Section 3, which introduces the answers to all research questions. Section 4 discusses the most relevant thoughts about learning-based accident prediction. Finally, the conclusions of this literature review are presented in Section 5.

## 2. Materials and Methods

The current study was performed using the guide for systematic reviews proposed by Kitchenham and others [10,11]. For this study, we have considered the following phases and activities: Planning the Review (Research Questions), Conducting the Review (Search Strategy, Study Selection, Study Quality Assessment, and Data Extraction), and Reporting the Review (Results).

### 2.1. Planning the Review
Research Questions

In this stage, we present seven research questions developed based on the goals of our research.

- RQ01. What are the data sources used by learning-based traffic accident prediction models?
- RQ02. Where were the datasets used by the prediction models extracted from?
- RQ03. What shortcomings are present in the prediction models?

- RQ04. What are the most common algorithms used by the prediction models?
- RQ05. What are the evaluation metrics used by the prediction models?
- RQ06. What is the performance obtained from the prediction models?
- RQ07. What percentages of the data are used by the models for training, validation, and testing?

### 2.2. Conducting the Review

#### 2.2.1. Search Strategy

The bibliographic databases and journal platforms used in this review were: Scopus, ACM Digital Library, IEEExplore, Springer Link, and Google Scholar. According to [12], Scopus and Web of Science provide a better quality of indexing and bibliographic records, at least in the computer science field. IEEExplore was picked out because it focuses exclusively on computer science, engineering, and electronics. ACM covers the area of computing and information technology. IEEExplore is considered one of the largest collections worldwide of technical literature. Finally, Springer Link was picked out because it contains many peer-reviewed journals and provides full-text access.

Based on the research questions presented, we extracted the following keywords: real-time, traffic accident prediction, learning, heterogeneous, data source, learning technique, algorithm, and evaluation metric. We added "predicting" and "forecast" to the keyword list as a synonym for prediction. We also developed a list of search strings combining the extracted keywords with the operators "AND" and "OR." We established three search strings (SS01, SS02, and SS03). SS01 is longer and more specific because it includes all the keywords and synonyms. SS02 does not include the keyword "real-time" from SS01, and SS03 that is less specific, does not include the keyword "heterogeneous" from SS02. This strategy implies that the results returned by each database or platform have duplicate items. Table 1 presents the search strings developed for this study and the search results.

**Table 1.** Search Results.

| Database Search Engine | ID | Command Search | Search Date | Total |
|---|---|---|---|---|
| Scopus | SS01 | ALL(real-time AND "traffic accident*" AND (predicti* OR forecast*) AND learning AND heterogeneous AND "data source*") | 1 April 2021 | 48 |
| | SS02 | ALL("traffic accident*" AND (predicti* OR forecast*) AND learning AND heterogeneous AND "data source*") | | 61 |
| | SS03 | ALL("traffic accident*" AND (predicti* OR forecast*) AND learning AND "data source*") | | 154 |
| | | | | 263 |
| ACM | SS01 | [All:real-time] AND [All:"traffic accident*"] AND [[All:predicti*] OR [All:forecast*]] AND [All:learning] AND [All:heterogeneous] AND [All:"data source*"] | 1 April 2021 | 10 |
| | SS02 | [All: "traffic accident*"] AND [[All:predicti*] OR [All:forecast*]] AND [All:learning] AND [All:heterogeneous] AND [All: "data source*"] | | 10 |
| | SS03 | [All: "traffic accident*"] AND [[All:predicti*] OR [All:forecast*]] AND [All:learning] AND [All:"data source*"] | | 13 |
| | | | | 33 |

**Table 1.** *Cont.*

| Database Search Engine | ID | Command Search | Search Date | Total |
|---|---|---|---|---|
| IEEExplore | SS01 | "Full Text & Metadata":real-time AND "Full Text & Metadata":"traffic accident*" AND ("Full Text & Metadata":predicti* OR "Full Text & Metadata":forecast*) AND "Full Text & Metadata":learning AND "Full Text & Metadata":heterogeneous AND "Full Text & Metadata":"data source*" | | 122 |
| | SS02 | "Full Text & Metadata":"traffic accident*" AND ("Full Text & Metadata":predicti* OR "Full Text & Metadata":forecast*) AND "Full Text & Metadata":learning AND "Full Text & Metadata":heterogeneous AND "Full Text & Metadata":"data source*" | 1 April 2021 | 136 |
| | SS03 | "Full Text & Metadata":"traffic accident*" AND ("Full Text & Metadata":predicti* OR "Full Text & Metadata":forecast*) AND "Full Text & Metadata":learning AND "Full Text & Metadata":"data source*" | | 360 |
| | | | | 618 |
| Springer | SS01 | real-time AND "traffic accident*" AND (predicti* OR forecast*) AND learning AND heterogeneous AND "data source*" | | 115 |
| | SS02 | "traffic accident*" AND (predicti* OR forecast*) AND learning AND heterogeneous AND "data source*" | 1 April 2021 | 145 |
| | SS03 | "traffic accident*" AND (predicti* OR forecast*) AND learning AND "data source*" | | 352 |
| | | | | 612 |
| Scholar | SS01 | real-time AND "traffic accident*" AND (predicti* OR forecast*) AND learning AND heterogeneous AND "data source*" | 1 April 2021 | 397 |
| | | | | 397 |

### 2.2.2. Study Selection

Some inclusion and exclusion criteria have been established to accomplish the study selection process.

- Inclusion criteria
    - IC01. Published in science, technology, and transportation journals and proceedings;
    - IC02. Peer-reviewed research papers;
    - IC03. Articles proposing traffic accident prediction models.
- Exclusion criteria
    - EC01. Published in health, psychology, or medical journals and proceedings;
    - EC02. Literature reviews, mapping studies, chapters in books, theses, technical reports, research proposals, lectures notes, or handbooks;
    - EC03. Published in preprint platforms;
    - EC04. Articles without full text;
    - EC05. Articles proposing traffic accident *detection* models.

### 2.2.3. Study Quality Assessment

In this stage, we defined the assessment questions used in the quality instrument. Additionally, we established two or three possible answers for each question and their scores. Thus, the answer "no" with 0 and "yes" is rated with 0.5 or 1.0 depending on the condition. We present the assessment questions and a short justification for them as follows.

The best way to evaluate a model is through the analysis of its evaluation metrics. Since some metrics are more robust and useful than others, having many of them helps to improve the model and its performance.

AQ01. Does the study present evaluation metrics

If the number of metrics = 1, the value is 0.5;
If the number of metrics > 1, the value is 1.0.

Determining the real causes of traffic accidents is complex because they depend on many factors. Thus, the success of such a prediction model lies in correlating different data sources.
AQ02. Does the prediction model correlate information from different data sources?

If the number of data sources = 1, the value is 0.5;
If the number of data sources > 1, the value is 1.0.

Proposing a prediction model by choosing one algorithm and calculating a metric is somewhat imprecise. This process requires an analysis of the model with several baseline algorithms to identify the best one based on indicators and metric values.
AQ03. Does the prediction model use different automatic learning algorithms?

If 0 < the number of algorithms ≤ 2, the value is 0.5;
If the number of algorithms > 2, the value is 1.0.

In general, the prediction models have to deal with high dimensionality and imbalance in information, poor handling of long-scale datasets, or insufficient capacities to process and analyze information. Our study also needs to know the challenges faced by traffic accident prediction models.
AQ04. Does the study present challenges that the prediction models must face?

If the study presents any challenge, the value is 1.0.

The correct handling of missing and out-of-range data will prevent the occurrence of a bias that invalidates the study. The following studies include missing data treatment in their proposals [13–17].
AQ05. Does the study include missing data treatment?

If the study includes any data missing treatment, the value es 1.0.

We established, as a selection criterion, that only if the sum of all five questions is greater than or equal to the value defined as the boundary for the first quartile, then the primary study is accepted; otherwise, it is rejected. This value corresponds to 2.5. The research community has widely accepted this selection criterion [11,18]. Table A1 presents the quality instrument and its results, and Figure 1 presents the phase of Conducting the Review. As observed, 1923 articles were found after performing the search strategy activity. Then, 778 duplicate articles were removed, giving a total of 1145 articles. Once the inclusion and exclusion criteria were applied, 1123 articles were excluded, giving a total of 22 articles. After performing the snowballing technique, 20 articles were added, giving a total of 42 articles. Finally, eight articles were rejected because they did not fulfill the quality criterion. Thus, the number of selected primary studies reached 34 papers. Table 2 presents the primary studies that were selected.

**Table 2.** Selected primary studies.

| ID | Authors | Title | Year | Type |
|----|---------|-------|------|------|
| PS01 | Hossain et al. [4] | A Bayesian network based framework for real-time crash prediction on the basic freeway segments of urban expressways | 2012 | Journal |
| PS02 | Wu et al. [19] | A Bayesian network model for real-time crash prediction based on selected variables by random forest | 2019 | Conference |
| PS03 | Ren et al. [20] | A Deep Learning Approach to the Citywide Traffic Accident Risk Prediction | 2018 | Conference |

**Table 2.** *Cont.*

| ID | Authors | Title | Year | Type |
|----|---------|-------|------|------|
| PS04 | Xu et al. [7] | A genetic programming model for real-time crash prediction on freeways | 2013 | Journal |
| PS05 | Wenqi et al. [21] | A model of traffic accident prediction based on convolutional neural network | 2017 | Conference |
| PS06 | Xiong et al. [22] | A New Framework of Vehicle Collision Prediction by Combining SVM and HMM | 2018 | Journal |
| PS07 | Lin et al. [9] | A novel variable selection method based on frequent pattern tree for real-time traffic accident risk prediction | 2015 | Journal |
| PS08 | Ozbayoglu et al. [23] | A real-time autonomous highway accident detection model based on big data processing and computational intelligence | 2016 | Conference |
| PS09 | Liu et al. [24] | A real-time explainable traffic collision inference framework based on probabilistic graph theory | 2021 | Journal |
| PS10 | Effati et al. [25] | A semantic-based classification and regression tree approach for modelling complex spatial rules in motor vehicle crashes domain | 2015 | Journal |
| PS11 | Bao et al. [26] | A spatiotemporal deep learning approach for citywide short-term crash risk prediction with multi-source data | 2019 | Journal |
| PS12 | Moosavi et al. [27] | Accident risk prediction based on heterogeneous sparse data: New dataset and insights | 2019 | Conference |
| PS13 | Yan et al. [28] | Crash prediction based on random effect negative binomial model considering data heterogeneity | 2020 | Journal |
| PS14 | Paikari et al. [5] | Data integration and clustering for real time crash prediction | 2014 | Conference |
| PS15 | Huang et al. [29] | Deep dynamic fusion network for traffic accident forecasting | 2019 | Conference |
| PS16 | Parra et al. [30] | Evaluating the Performance of Explainable Machine Learning Models in Traffic Accidents Prediction in California | 2020 | Conference |
| PS17 | Yuan et al. [31] | Hetero-ConvLSTM: A deep learning approach to traffic accident prediction on heterogeneous spatio-temporal data | 2018 | Conference |
| PS18 | Huang et al. [32] | Highway crash detection and risk estimation using deep learning | 2018 | Journal |
| PS19 | Park et al. [33] | Highway traffic accident prediction using VDS big data analysis | 2016 | Journal |
| PS20 | Chen et al. [34] | Learning deep representation from big and heterogeneous data for traffic accident inference | 2016 | Conference |
| PS21 | Golovnin et al. [35] | Operational forecasting of road traffic accidents via neural network analysis of big data | 2020 | Journal |
| PS22 | Wang et al. [36] | Predicting Crashes on Expressway Ramps with Real-Time Traffic and Weather Data | 2015 | Journal |
| PS23 | Yuan et al. [37] | Predicting traffic accidents through heterogeneous urban data: A case study | 2017 | Conference |

**Table 2.** *Cont.*

| ID | Authors | Title | Year | Type |
|---|---|---|---|---|
| PS24 | Effati et al. [38] | Prediction of Crash Severity on Two-Lane, Two-Way Roads Based on Fuzzy Classification and Regression Tree Using Geospatial Analysis | 2015 | Journal |
| PS25 | Wang et al. [8] | Real-time crash prediction for expressway weaving segments | 2015 | Journal |
| PS26 | Basso et al. [6] | Real-time crash prediction in an urban expressway using disaggregated data | 2018 | Journal |
| PS27 | Zhang et al. [39] | RiskCast: Social sensing based traffic risk forecasting via inductive multi-view learning | 2019 | Conference |
| PS28 | Chen et al. [40] | SDCAE: Stack Denoising Convolutional Autoencoder Model for Accident Risk Prediction Via Traffic Big Data | 2018 | Conference |
| PS29 | Zhou et al. [41] | Stack ResNet for Short-term Accident Risk Prediction Leveraging Cross-domain Data | 2019 | Conference |
| PS30 | Dong et al. [42] | Support vector machine in crash prediction at the level of traffic analysis zones: Assessing the spatial proximity effects | 2015 | Journal |
| PS31 | Zhu et al. [43] | TA-STAN: A Deep Spatial-Temporal Attention Learning Framework for Regional Traffic Accident Risk Prediction | 2019 | Conference |
| PS32 | Yu et al. [44] | Traffic accident prediction based on deep spatio-temporal analysis | 2019 | Conference |
| PS33 | Sharma et al. [45] | Traffic accident prediction model using support vector machines with Gaussian kernel | 2016 | Conference |
| PS34 | Al Mamlook et al. [46] | Utilizing Machine Learning Models to Predict the Car Crash Injury Severity among Elderly Drivers | 2020 | Conference |

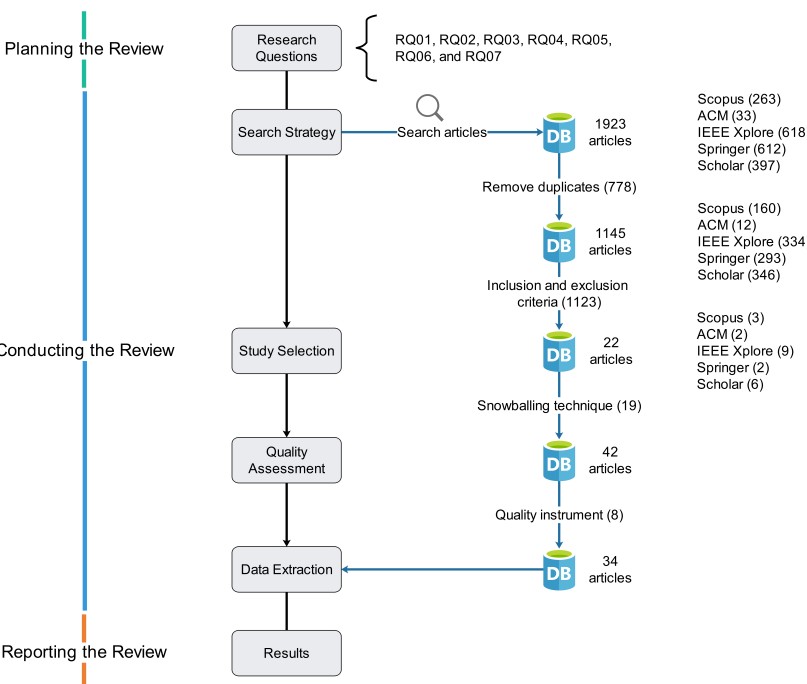

**Figure 1.** The review process.

2.2.4. Data Extraction

We designed four data collection forms to record the selected primary studies' information. The data collection forms proposed for this section are shown in Tables 3, 4, A2 and A3. The design of them was based on addressing the research questions. Thus, Table A2 was designed to answer RQ01, Table A3 to answer RQ02, Table 3 to answer RQ04, and Table 4 to answer RQ05, RQ06, and RQ07. Table A2 includes the primary study ID, the data sources (vehicle data, driver's data, weather and light conditions, traffic accidents, traffic flow, traffic events, road infrastructure, taxi trips, points of interest, and others), two categories to refer to the data type, and a list of variables of features of each data source. Table A3 includes the primary study ID, the datasets, services, or simulators. Table 3 includes the primary study ID, the algorithm or algorithms used on the model, and the groups to which those belong [47,48]. Finally, Table 4 includes the primary study ID, some evaluation metrics, percentages of data used for training, validation, and testing, and the algorithms used by models to compare their performance. The generated data will be presented in the "Results" section and analyzed and interpreted in the "Discussion".

**Table 3.** Classification of algorithms by category.

| | Algorithms | | |
| ID | Algorithms/Probability Models | Ranking/Variables Selection | Categories |
| --- | --- | --- | --- |
| PS01 | Bayesian Belief Net | Random Multinomial Logit | Classification |
| PS02 | Bayesian Network | Random Forest | Classification/Ensemble |
| PS03 | Long Short-Term Memory Neural Network | | Neural Networks |
| PS04 | Genetic Programming | Random Forest | Evolutionary Computation/ Ensemble |
| PS05 | Convolutional Neural Network | | Neural Networks |
| PS06 | Support Vector Machine | | Classification |
| PS07 | Bayesian Network | Frequent Pattern Tree/ Random Forest | Classification |
| PS08 | K-Nearest Neighbor/Regression Tree/Feed Forward Neural Network | | Classification/Ensemble/ Neural Networks |
| PS09 | Bayesian Network | | Classification |
| PS10 | Ontology-based Classification and Regression Tree | | Classification/Regression |
| PS11 | Convolutional Long Short-Term Memory Neural Network | | Neural Networks |
| PS12 | Deep Neural Network | | Neural Networks |
| PS13 | Negative Binomial/Random Negative Binomial | | Probability Distributions |
| PS14 | Bayesian Network | | Classification |
| PS15 | Multilayer Perceptron | | Neural Networks |
| PS16 | Gradient Boosting | | Ensemble |
| PS17 | Convolutional Long Short-Term Memory | | Neural Networks |
| PS18 | Convolutional Neural Network | | Neural Networks |
| PS19 | K-Means/Logistic Regression | | Clustering/Classification |

**Table 3.** *Cont.*

| ID | Algorithms | | |
| | **Algorithms/Probability Models** | **Ranking/Variables Selection** | **Categories** |
|---|---|---|---|
| PS20 | Stack Denoise Autoencoder | | Neural Networks |
| PS21 | Rumelhart Multilayer Perceptron | | Neural Networks |
| PS22 | Bayesian Logistic Regression | Random Forest | Classification/Ensemble |
| PS24 | Fuzzy Classification and Regression Tree | | Classification/Regression |
| PS25 | Bayesian Logistic Regression | Random Forest | Classification/Ensemble |
| PS26 | Support Vector Machine/Logistic Regression | Random Forest | Classification |
| PS27 | Multi-view Learning | | Not available |
| PS28 | Stack Denoise Convolutional Autoencoder | | Neural Networks |
| PS29 | Convolutional Neural Network | | Neural Networks |
| PS30 | Support Vector Machine with radial-basis function | | Classification |
| PS31 | Deep Learning | | Neural Networks |
| PS32 | Long Short-Term Memory Neural Network and Fully Connected Network | | Neural Networks |
| PS33 | Support Vector Machine with Gaussian kernel | | Classification |
| PS34 | Light Gradient Boosting Machine | | Ensemble |

**Table 4.** Performance of models.

| ID | Evaluation Metrics | | | | | | | | | Percentage of Data | | | Compared with |
| | **MAE** | **MRE** | **RMSE** | **MSE** | **PAR %** | **FPR %** | **TPR %** | **F1** | **AUC** | **Train. %** | **Valid. %** | **Test. %** | |
|---|---|---|---|---|---|---|---|---|---|---|---|---|---|
| PS01 | | | | | 66.00 | 20.00 | | | | | | | Not available |
| PS02 | | | | | | 16.07 | 70.46 | | | | | | K-Nearest Neighbor, Support Vector Machine, and Logistic Regression |
| PS03 | 0.014 | | 0.034 | 0.001 | | | | | | | | | LASSO and Ridge Regression, Support Vector Regression, Decision Tree Regression, Random Forest Regression, Multilayer Perceptron, and Autoregressive Moving Average |
| PS04 | | | | | | | 75.40 | | | | | | Binary Logistic Regression |
| PS05 | | | | | 78.50 | | | | | 60.0 | | 40.0 | Backpropagation Network |
| PS06 | | | | | 96.70 | | | | | 75.0 | | 25.0 | Not available |
| PS07 | | | | | | 38.16 | 61.11 | | | 80.0 | | 20.0 | K-Nearest Neighbor |

**Table 4.** *Cont.*

| ID | Evaluation Metrics | | | | | | | | | Percentage of Data | | | Compared with |
|----|-----|-----|------|-----|------------|------------|------------|-----|-----|-------------|-------------|------------|---------------|
| | MAE | MRE | RMSE | MSE | PAR % | FPR % | TPR % | F1 | AUC | Train. % | Valid. % | Test. % | |
| PS08 | | | | | 99.79 | | 42.86 | | | | | | K-Nearest Neighbor/Regression Tree |
| PS09 | | | | | | | 0.887 | 0.813 | | var. | | var. | Artificial Neural Network, Bayesian Regression, and Naive Bayes |
| PS10 | | | 0.267 | | | | 0.818 | 0.803 | 0.807 | 70.0 | | 30.0 | Not available |
| PS11 | 0.023 | | | 0.019 | 81.58 | 0.34 | | | | | | | Convolutional Neural Network, Long Short-Term Memory Neural Network, Artificial Neural Network, and Gradient Boosting Regression Tree |
| PS12 | | | | | | | | 0.590 | | 83.0 | | 17.0 | Logistic Regression and Gradient Boosting |
| PS13 | 2.520 | | 0.290 | | | | | | | | | | Negative Binomial |
| PS14 | | | | | 76.0 | | | | | | | | Not available |
| PS15 | | | | | | | | 0.681 | 0.786 | | | | Support Vector Regression, Logistic Regression, Deep Neural Network, Long-Short Term Memory, and Recurrent Neural Network |
| PS16 | | | | | 78.00 | | 73.00 | 0.740 | | 70.0 | 30.0 | | Decision Tree and Random Forest |
| PS17 | | | 0.116 | 0.013 | | | | | | 79.0 | 9.0 | 12.0 | Least Squares Linear Regression, Decision Tree Regression, Deep Neural Network, Fully Connected Long Short-Term Memory, and Convolutional Long Short-Term Memory |
| PS18 | | | | | 77.34 | | | 0.765 | | 80.0 | | 20.0 | Convolutional Neural Network |
| PS19 | | | | | 76.35 | | 40.83 | | | 75.0 | | | Logistic Regression and Support Vector Machine |
| PS20 | 0.96 | 0.39 | 1.0 | | | | | | | 80.0 | | 20.0 | Decision Tree, Logistic Regression, and Support Vector Machine |
| PS21 | | | | | | | | | 0.90 | | | | Not available |
| PS22 | | | | | 90.49 | | 90.40 | | 0.971 | 70.0 | 30.0 | | Not available |
| PS23 | | | | | 95.12 | | 0.868 | 0.898 | 0.961 | var. | | var. | Support Vector Machine, Decision Tree, and Random Forest |
| PS24 | | | | | 79.12 | | 0.68 | | | | | | Classification and Regression Tree and Support Vector Machine |
| PS25 | | | | | 69.80 | | 67.60 | | | 70.0 | 30.0 | | Not available |
| PS26 | | | | | | | 75.03 | | | 80.0 | 20.0 | | Support Vector Machine |
| PS27 | 1.569 | | | | | | | | | | | | Linear Regression /Ridge Regression/Multilayer Perceptron |
| PS28 | 0.092 | 0.796 | | | | | | | | 80.0 | 20.0 | | Logistic Regression, Random Forest, Decision Tree, Linear Regression, and Stack Denoise Autoencoder |

**Table 4.** *Cont.*

| ID | Evaluation Metrics | | | | | | | | | Percentage of Data | | | Compared with |
|----|-----|-----|------|-----|-----------|-----------|-----------|-----|-----|------------|-------------|------------|----------------|
|    | MAE | MRE | RMSE | MSE | PAR % | FPR % | TPR % | F1 | AUC | Train. % | Valid. % | Test. % |  |
| PS29 |  |  | 0.40 | 0.16 | 88.89 |  |  |  |  | 87.0 |  | 13.0 | Auto-Regressive Integrated Moving Average, and Convolutional Long Short-Term Memory Neural Network |
| PS30 |  |  |  |  | 81.3 |  |  |  |  | 80.0 |  | 20.0 | Support Vector Machine with linear |
| PS31 | 0.0082 |  | 0.0131 | 0.0001 |  |  |  |  |  | 67.0 | 11.0 | 22.0 | Linear Regression, Long Short-Term Memory Neural Network, Denoising Auto-Encoder, XG-Boost, and Seq2Seq |
| PS32 |  |  | 0.444 |  | 0.723 |  | 0.773 | 0.736 |  | 70.0 | 10.0 | 20.0 | Logistic Regression, Least Absolute Shrinkage and Selection Operator, Support Vector Machine, and Decision Tree |
| PS33 |  |  |  |  | 94.0 |  |  |  |  | 70.0 | 20.0 | 10.0 | Multilayer Perceptron and Support Vector Machine with poly kernel |
| PS34 |  |  |  |  | 87.54 |  | 0.814 | 0.837 |  | 80.0 |  | 20.0 | Logistic Regression, Decision Tree, Random Forest, and Naive Bayesian |

## 3. Results

### 3.1. Study Overview

Considering the year and the type of publication (Table 2), from 34 selected studies, 19 of them are articles from journals and 15 of them from conferences. The years in which more papers were published were 2015, 2018, and 2019. The answers to our research questions are presented as follows.

### 3.2. RQ01

The prediction models use the following data sources: vehicle data, driver's data, weather conditions, light conditions, traffic accidents, traffic flow, traffic events, road infrastructure, taxi trips, points of interest, and population. The most common data sources are weather conditions, traffic accidents, traffic flow, and road infrastructure. Meanwhile, driver's data, light conditions, and taxi trips are the least common. Based on Table A2, the attributes contained in each data source are presented as follows.

- Vehicle data: identifier, time, location, type, speed, condition, seat belt, pick up and pick off time;
- Driver's data: age, gender, education level, collision factors (sleepiness and boredom), and involvement of alcohol and drugs;
- Weather conditions: sun, cloud, rain, snow, fog, sleet, crosswind, sand, dawn, dusk, visibility, temperature, precipitation, snowfall, pressure, wind speed, humidity, hail, storm, wind direction, and dew point;
- Light conditions: headlights, streetlights, sunlight, and night light;
- Traffic accidents: vehicles involved, collision type, collision description, the direction of the road, number of killed or injured people, severity, human situation, number of property damage only collisions, number of collisions with casualties and dead, presence of traffic objects, road segment, event type, security level, collision month, vehicle failure, police report, and origin of the collision;
- Traffic flow: vehicle speed according to radar, number of vehicles, occupancy, average speed, annual average daily traffic, driving direction, and lane identifier;

- Traffic events: closures, constructions, broken vehicles, collisions, congestion, and blocked lanes;
- Road infrastructure: geometric characteristics (road length, road shape, road alignment, road type, number of lanes, horizontal curve radius, width of shoulder, slope, tunnel, imperfections, intersections, entrance and exit ramp, and speed limits), and road signs (warning, priority, information, facilities, and service);
- Taxi trips: pick-up and drop-off timestamp, pick-up and drop-off location, trip distance, payment information, taxi zones, and taxi speed;
- Points of interest: place, category, and location;
- Population: *not shown;
- Other: topographic map, digital elevation map, land use, satellite images, the area size of census blocks, special calendar dates, geographical area, trip survey, and bike trip.

### 3.3. RQ02

The prediction models are fed with data collected from open and government platforms, others from Internet services, and even others with simulators' data. According to Table A3, the platforms, Internet services, and simulators used by the models to collect data are presented as follows.

- Open platforms: Kaggle and Open Data;
- Government platforms: Institutions of statistics and census, geographical and meteorological organizations, and departments of police and transportation;
- Internet services: MapQuest Traffic, Microsoft Bing Map Traffic, The Weather Channel, Weather Underground, Google Earth Satellite Image, and Twitter;
- Simulators: AIMSUN, VISSIM, PreScan, and Paramics Discovery;
- Applications: Intelligent Transportation Systems (ITS) and Real-Time Monitoring Systems;
- Others: Questionnaires.

### 3.4. RQ03

Considering that "no model is perfect", the prediction models present at least some of the following shortcomings.

- Non-inclusion of spatial heterogeneity within the zones of study;
- Information imbalance (the amount of useless data is greater than useful data) because most data are non-accident related;
- Insufficient capacities to process and analyze an enormous amount of data;
- Poor handling of long-scale datasets. It is not practical to work with huge amounts of raw data; therefore, it is necessary to select relevant features to be extracted. If this selection is not made adequately, the generated models will not work correctly;
- Not having enough related information to train and test the models (e.g., it is essential to have information about traffic accidents and normal traffic conditions from the same segment).

### 3.5. RQ04

The most common algorithms among prediction models in order of occurrence are Neural Networks (Long Short-Term Memory NN, Convolutional NN, Deep NN, and Feed Forward NN), Support Vector Machine, and Bayesian Networks. According to Figure 2, 30% of prediction models use some variants of Neural Networks, 15% of them use Support Vector Machine, and 12% use Bayesian Networks. Regarding ranking and selection variables/features, the most common algorithm is Random Forest. The categories to which those algorithms belong are Neural Networks, Classification, and Ensemble. Finally, the most common algorithms used by models to compare their performance are Logistic Regression, Support Vector Machine, Decision Tree, and some variants of Neural Networks. Their categories are *Classification* and *Neural Networks*.

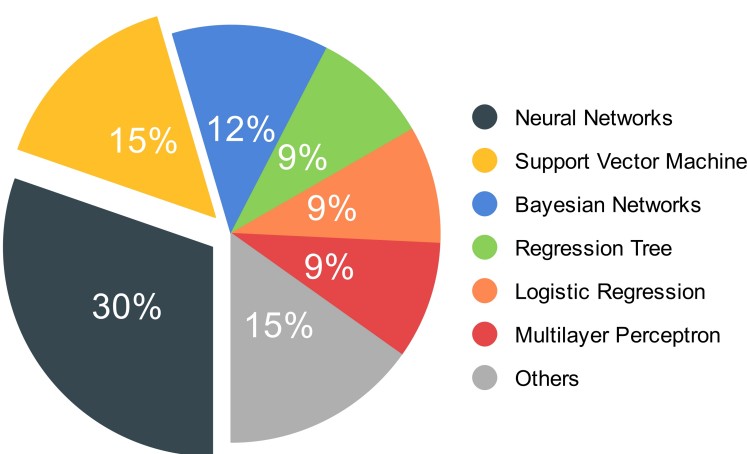

**Figure 2.** Distribution of algorithms in traffic accident prediction models.

*3.6. RQ05*

The evaluation metrics used by authors are:

- For classification problems: Prediction Accuracy Rate (PAR)/Accuracy, True Positive Rate (TPR)/Sensitivity/Recall, False Positive Rate (FPR)/Fall-Out, F1 Score, and Area Under Curve (AUC);
- For regression problems: Mean Absolute Error (MAE), Mean Relative Error (MRE), Root Mean-Square Error (RMSE), and Mean Squared Error (MSE).

Of all these metrics, the more commonly used are:

- For classification: PAR, TPR, and F1 Score;
- For regression: RMSE and MAE.

*3.7. RQ06*

The prediction models obtained the results presented as follows. Figure 3 shows the dispersion of values of evaluation metrics.

- Accuracy (%):
  PS01 ≫ 66.00      PS05 ≫ 78.50      PS06 ≫ 96.70      PS08 ≫ 99.79      PS11 ≫ 81.58
  PS14 ≫ 76.00      PS16 ≫ 78.00      PS18 ≫ 77.34      PS19 ≫ 76.35      PS23 ≫ 95.12
  PS24 ≫ 79.12      PS25 ≫ 69.80      PS29 ≫ 88.89      PS30 ≫ 81.30      PS33 ≫ 94.00
  PS34 ≫ 87.54
- Sensitivity (%):
  PS02 ≫ 70.46      PS04 ≫ 75.40      PS07 ≫ 66.11      PS26 ≫ 75.03
- F1 score:
  PS09 ≫ 0.813      PS10 ≫ 0.803      PS12 ≫ 0.590      PS15 ≫ 0.681
- Root Mean-Square Error:
  PS03 ≫ 0.034      PS13 ≫ 0.290      PS17 ≫ 0.116      PS32 ≫ 0.444
- Mean Absolute Error:
  PS20 ≫ 0.960      PS27 ≫ 1.569      PS28 ≫ 0.092      PS31 ≫ 0.008
- Area Under Curve:
  PS21 ≫ 0.900

According to Figure 3, there are four groups of values for PAR, F1, AUC, and MSE. All PAR values range from 0.65 to 1.0 (65% to 100%). Similarly, all F1 and AUC values range from 0.58 to 0.90 and 0.80 and 0.97, respectively. Additionally, all MSE values are located under 0.17. These ranges could be seen as a reference for new models that use these evaluation metrics. By contrast, the values of the rest of metrics are so dispersed that it is not possible to identify group of values to serve as references.

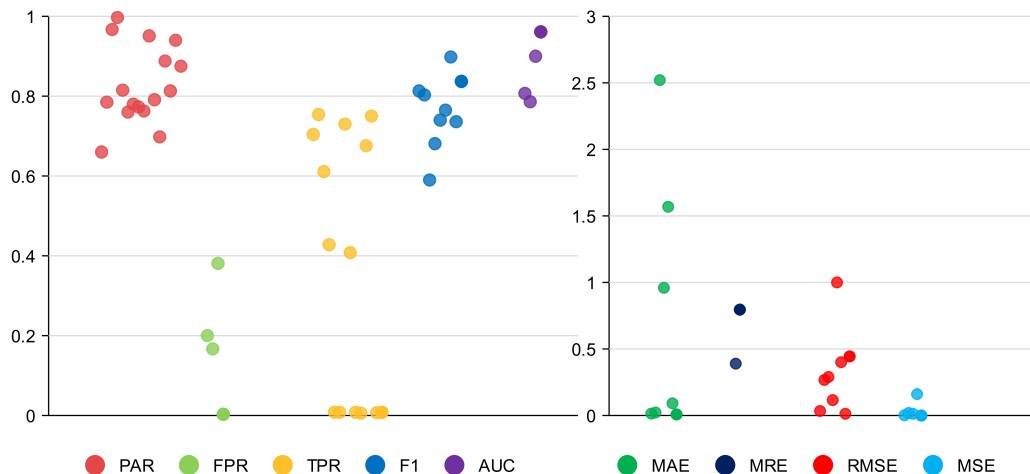

**Figure 3.** Dispersion of values of evaluation metrics.

*3.8. RQ07*

Most models only use data for training and testing; however, a few models also use data for validation. The percentages established by the models are as follows:

- For training: [60.0–83.0]%
- For validation: [9.0–30.0]%
- For testing: [10.0–40.0]%

Even there are models in which those percentages are variable and defined dynamically. The most common split configuration among proposals is 80% for training and 20% for testing.

## 4. Discussion

Below, we mention some thoughts presented in the articles to analyze and consider for future research. For instance, traffic accidents are not fortuitous events but events caused by conditions that occur in space and time and under certain circumstances [30]. According to [32], unfavorable traffic characteristics, adverse weather conditions, and driver distraction may lead to a crash. Additionally, the most significant factors on crash severity are vehicle failures, not wearing the seat belt, and unfavorable weather conditions [38]. Meanwhile, others assert that driving drunk and at high speed are serious factors in traffic accidents [45], and the wet pavement is one condition that increases the accident rate significantly [8]. Finally, the situation that causes the highest probability of suffering a traffic accident is the aggressive driving behavior after unusual congestion to recover the time lost [6]. For their part, the authors of [25] determined as follows: high speed is one of the most recurrent causes among fatal vehicle crashes; the traffic during morning peak and the first days of the week increase the risk of property-damage-only crashes; additionally, slopes and proximity to curves are the main road geometry factors that lead to fatal crashes; high speed and proximity to curves are the main causes of fatal-injury type crashes; faulty windshield wipers in rainy weather conditions and not wearing seat belt among young people are the most important causes of injury crashes; and, finally, driving at night without caution during rainy weather conditions increase the risk of property-damage-only crashes.

About performance, models based on Deep Neural Networks reduce their accuracy, precision, and F1 score as the learning data size increases [37]. Additionally, the performance of a Support Vector Machine model depends on the learning process, so future efforts must focus on tuning the scale of parameter values and kernel functions selection [42]. Finally, the authors of [26] assert that the performance of the prediction models decreases as the spatio-temporal resolution of the prediction task increases. Regarding features, incorporating more features into the model does not always improve its performance [44]. Meanwhile, ref. [45] asserts that a lesser number of features affect the performance of a neural network. Finally, and according to [37], removing features from models based on

Decision Tree or Random Forest has an enormous impact, but slight in models based on Deep Neural Networks.

Some authors propose some recommendations; for instance, splitting data into pieces to send them to compute nodes can make the computational time much lower, which would benefit the handling of social media data [24]. For their part, ref. [19] suggests that the threshold used to separate different states (crash/non-crash) must compensate the values of True Positive Rate and False Positive Rate, and also that the optimal threshold may be found by comparing the performance of different thresholds. Finally, the authors of [5] proposes that the outcomes of a real-time traffic accident prediction model are shown through a variable message sign or transmitted between vehicles using a connected vehicular system.

Despite the advantages that simulators offer at present, this mechanism of data generation has not been received as expected. In fact, there is a clear trend in prediction models about using real data instead of simulated ones. From the results, we could remark that only 1 out of 10 models use data generated by simulators. Because traffic accidents are events caused by a group of conditions that are not always the same and take place in space and time and under certain circumstances, it can be suspected that the authors prefer less controlled scenarios than those provided by simulators to generate data. Moreover, it was noted that there are both static and variable data. Static data, such as most driver data, road infrastructure, points of interest, or satellite images could be used to build a base model. In contrast, data that vary over time, such as traffic accidents, weather conditions, or traffic flow, could be used to adjust the model.

The human factor is the leading cause of traffic accidents [49,50], and the most common human factor (contributing or principal) is inattention while driving because of overloading attention, distraction, or monotonous driving [51]. According to [46], young people are more susceptible than adults to suffer a traffic accident; male drivers are more involved in traffic incidents than female drivers, and female drivers are more susceptible than male drivers to suffering severe injuries. It is clear that the human factor influences and plays an essential role in the occurrence and severity of traffic accidents. This affirmation is confirmed in the Global Status Report on Road Safety. It establishes that factors associated with road user behavior, such as speeding and drink-driving, are two of the key risk factors to be considered and reinforced within the legislation of countries to prevent deaths and injuries due to traffic accidents. Some countries, especially high-income ones, have reduced the number of deaths and injuries by adopting policies for all the key risk factors [1]. Although we have improved much in the prevention of traffic accidents, it is clear that we must now focus on the field of the prediction of traffic accidents. In the context of our research, we could notice that very few models use driver's data, although the human factor is one of the leading causes of traffic accidents. We believe that this may be due to the non-availability of this type of information.

Considering that the prediction models are generally fed with real data, the authors have resorted primarily to governmental institutions related to transportation or related areas and secondly to Internet services. The information collected from government platforms is mainly related to traffic accidents, traffic flow, and road infrastructure. The information collected from government platforms is mainly related to traffic accidents, traffic flow, and road infrastructure. Internet services provide information mainly related to weather and light conditions and traffic events. Most Internet services (MapQuest Traffic, Microsoft Bing Map Traffic, or Twitter, among others) provide APIs that can be integrated into the model to establish real-time or deferred information channels.

One of the most challenging issues for the traffic accident prediction models is to count on a real-time solution. According to some authors [4,9,33,37], the development of a real-time decision-making tool to avoid traffic accidents is completely viable as soon as shortcomings such as the non-integration of spatial heterogeneity, the incorrect handling of long-scale datasets, the improper handling of unique data properties, the information imbalance, and the lack of related information, are resolved. The correct handling of long-scale datasets requires feature extraction and imbalance correction. First of all, it is not

practical to work with a huge amount of raw data, therefore to handle adequately large datasets, it is necessary to extract essential features such as weather, type of environment (for instance, rural highway vs. urban street), road conditions, speed limit, type of traffic, driver data, and type of vehicles [26]. Additionally, the accident-related data are less frequent than the non-accident-related information. Therefore, the datasets are imbalanced, and a predicting model has to be built to correct this situation [32].

It is also essential to consider which characteristics are time-sensitive, time-insensitive, and related to spatial heterogeneity. Time-insensitive data are fully connected, and spatial heterogeneity is a trainable component. It would be possible to obtain a somewhat generalized solution trained for different scenarios starting with a common base that considers this feature differentiation type [52]. These data-handling strategies could make it possible to obtain a real-time prediction which is the next big challenge for this research area.

The high-dimensionality problem may be solved using data processing techniques to derive relevant features through methods, such as clustering, chi-square, Minimum-Redundancy-Maximum-Relevance (mRMR), and predictor importance, among others. Some authors, such as [28], have worked on this strategy for dimensionality reduction using clustering, but other pre-processing techniques could also be tested.

Regarding algorithms, we were able to identify two stages for which machine learning algorithms were assigned. The pre-processing stage includes the tasks of ranking and selecting features, while the classification stage includes the selection of the model. For pre-processing, the most common algorithm is Random Forest; and, for classification, the most common algorithms are some variants of Neural Networks (Long Short-Term Memory NN, Convolutional NN, Deep NN, and Feed Forward NN). This algorithm selection is consistent with the fact that deep learning models applied in the area of Traffic Accident Prediction are becoming more popular. Most authors use shallow learning algorithms as baseline algorithms to compare the performance of their models based on neural networks. This tendency marks a path for research in learning-based accident prediction.

The metrics more commonly used for classification problems are accuracy, sensitivity, and F1 Score; meanwhile, for regression problems, Mean Absolute Error and Root Mean-Square Error. However, there is such a diversity of experimental design, data volume, and structure used in the various studies that it is difficult to compare results using simply evaluation metrics. Not to mention that some proposals present non-normalized values for their evaluation metrics. The datasets are typically unbalanced, and performance must be understood in a contextualized way. Therefore, to compare models to find the one with the best performance is not necessarily real because the results are not completely comparable among the studies.

Although there is no precise rule to split data for training, validation, and testing, a tacit agreement establishes an approximate data split configuration. From the analysis, we could establish that a higher percentage (more than 50%) of data are used for training and a lower percentage (less than 50%) for testing. The most common data split configuration among proposals is 80% for training and 20% for testing. Some models even establish a low percentage of data for validation. It was noted that there is no evidence or justification for splitting data in one way or the other or whether such a data split configuration could improve the performance of the models. Because of this drawback, we could suggest using data splitting methods (e.g., SPlit [53]) instead of splitting randomly to obtain the optimal configuration.

## 5. Conclusions

The elaboration of this work has made it possible to present a review of the research done so far on learning-based traffic accident prediction. Some of the most important points to be considered are as follows.

The development of prediction models in real-time is viable as soon as issues, such as the efficient use of large-scale datasets, the integration of spatial heterogeneity, and the solution for high dimensionality in data, are resolved. In this context, some solutions for these issues are presented as follows. The efficient handling of large-scale datasets

may be solved using feature extraction and imbalance correction; meanwhile, the high dimensionality in data may be solved using data processing techniques.

There is a trend about using real data generated by less controlled scenarios (as in real life) instead of data generated by simulators. Thus, authors have opted to correlate real data usually collected from open and government platforms with information from Internet services. Additionally and through APIs, real-time or deferred information channels may be integrated into the model.

The performance of a prediction model depends largely on the quality of data, the set of algorithms, among others, but also depends on the data split configuration. Despite not having with specific and exact mechanism is fundamental to count on a strategy to establish the correct percentages of data for training, validation, and testing. Using splitting methods instead of splitting randomly to obtain the optimal configuration may be an option.

Future research must point to developing prediction models using deep learning (a combination of supervised and unsupervised learning techniques) and be focused on using data sources little used in traffic accident predictions (driver's data and pedestrian mobility).

**Author Contributions:** Conceptualization, P.M.; methodology, P.M.; writing—original draft preparation, P.M.; writing—review and editing, P.M., Á.L.V.C. and M.H.-Á.; supervision, Á.L.V.C. and M.H.-Á. All authors have read and agreed to the published version of the manuscript.

**Funding:** This research was funded by Escuela Politécnica Nacional grant number PIS 20-02 (Emergent System based on acquisition, processing, and response agents for management of vehicle accident rate using artificial intelligence techniques).

**Acknowledgments:** Our recognition to VIIV (Vicerrectorado de Investigación, Innovación y Vinculación) of Escuela Politécnica Nacional.

**Conflicts of Interest:** The authors declare no conflict of interest. The funders had no role in the design of the study; in the collection, analyses, or interpretation of data; in the writing of the manuscript, or in the decision to publish the results.

## Abbreviations

The following abbreviations are used in this manuscript:

| | |
|---|---|
| PAHO | The Pan American Health Organization |
| WHO | The World Health Organization |
| GSRRS | Global Status Report on Road Safety |
| SVM | Support Vector Machine |
| HMM | Hidden Markov Model |
| LSTM | Long Short-Term Memory |
| VDS | Vehicle Detection Sensor |
| DBSCAN | Density-Based Spatial Clustering of Applications with Noise |
| NN | Neural Network |
| LASSO | Least Absolute Shrinkage and Selection Operator |

## Appendix A

**Table A1.** Quality instrument.

| Title | AQ01 | AQ02 | AQ03 | AQ04 | AQ05 | Total |
|---|---|---|---|---|---|---|
| A Bayesian network based framework for real... [4] | 1.0 | 1.0 | 0.5 | 1.0 | 0.0 | **3.5** |
| A Bayesian network model for real-time crash... [19] | 1.0 | 1.0 | 0.5 | 1.0 | 0.0 | **3.5** |
| A crash-prediction model for multilane roads [54] | 0.0 | 1.0 | 1.0 | 0.0 | 0.0 | 2.0 |
| A deep learning approach to the Citywide... [20] | 1.0 | 0.5 | 1.0 | 1.0 | 0.0 | **3.5** |
| A genetic programming model for real-time crash... [7] | 0.5 | 1.0 | 0.5 | 1.0 | 0.0 | **3.0** |

**Table A1.** *Cont.*

| Title | AQ01 | AQ02 | AQ03 | AQ04 | AQ05 | Total |
|---|---|---|---|---|---|---|
| A model of traffic accident prediction based on. . . [21] | 0.5 | 1.0 | 0.5 | 1.0 | 0.0 | **3.0** |
| A New Framework of Vehicle Collision. . . [22] | 0.5 | 1.0 | 0.5 | 1.0 | 0.0 | **3.0** |
| A novel variable selection method based on. . . [9] | 1.0 | 1.0 | 0.5 | 1.0 | 1.0 | **4.5** |
| A real-time autonomous highway accident. . . [23] | 1.0 | 0.5 | 1.0 | 1.0 | 0.0 | **3.5** |
| A real-time explainable traffic collision inference. . . [24] | 1.0 | 1.0 | 1.0 | 1.0 | 0.0 | **4.0** |
| A rear-end collision prediction scheme based on. . . [55] | 0.5 | 0.5 | 0.5 | 0.0 | 0.0 | 1.5 |
| A semantic-based classification and regression. . . [25] | 1.0 | 1.0 | 0.5 | 1.0 | 0.0 | **3.5** |
| A spatiotemporal deep learning approach for. . . [26] | 1.0 | 1.0 | 1.0 | 1.0 | 0.0 | **4.0** |
| Accident risk prediction based on heterogeneous. . . [27] | 1.0 | 1.0 | 1.0 | 1.0 | 0.0 | **4.0** |
| Crash prediction based on random effect. . . [28] | 1.0 | 1.0 | 0.5 | 1.0 | 0.0 | **3.5** |
| Data integration and clustering for real time crash. . . [5] | 0.0 | 1.0 | 0.5 | 1.0 | 0.0 | 2.5 |
| Deep dynamic fusion network for traffic accident. . . [29] | 1.0 | 1.0 | 1.0 | 1.0 | 0.0 | **4.0** |
| Evaluating the Performance of Explainable. . . [30] | 1.0 | 0.5 | 1.0 | 1.0 | 0.0 | **3.5** |
| Hetero-ConvLSTM: A deep learning approach to. . . [31] | 1.0 | 1.0 | 1.0 | 1.0 | 0.0 | **4.0** |
| Highway crash detection and risk estimation. . . [32] | 1.0 | 1.0 | 0.5 | 1.0 | 1.0 | **4.5** |
| Highway traffic accident prediction using VDS. . . [33] | 1.0 | 1.0 | 1.0 | 1.0 | 0.0 | **4.0** |
| Intelligent algorithm in a smart wearable device. . . [56] | 0.0 | 1.0 | 0.5 | 0.0 | 0.0 | 1.5 |
| Learning deep representation from big and. . . [34] | 1.0 | 1.0 | 1.0 | 1.0 | 0.0 | **4.0** |
| Operational forecasting of road traffic accidents. . . [35] | 0.5 | 1.0 | 0.5 | 1.0 | 0.0 | **3.0** |
| Predicting crashes on expressway ramps with. . . [36] | 1.0 | 1.0 | 0.5 | 1.0 | 0.0 | **3.5** |
| Predicting motor vehicle crashes using Support. . . [52] | 0.0 | 0.5 | 0.5 | 1.0 | 0.0 | 2.0 |
| Predicting traffic accidents through. . . [37] | 1.0 | 1.0 | 1.0 | 1.0 | 1.0 | **5.0** |
| Prediction of Crash Severity on Two-Lane, Two. . . [38] | 1.0 | 1.0 | 1.0 | 1.0 | 1.0 | **5.0** |
| Real-time crash prediction for expressway. . . [8] | 1.0 | 1.0 | 0.5 | 1.0 | 0.0 | **3.5** |
| Real-time crash prediction in an urban. . . [6] | 0.5 | 1.0 | 1.0 | 1.0 | 0.0 | **3.5** |
| RiskCast: Social sensing based traffic risk. . . [39] | 0.5 | 1.0 | 1.0 | 0.0 | 0.0 | 2.5 |
| Real-time estimation of accident likelihood for. . . [57] | 0.0 | 1.0 | 0.5 | 0.0 | 0.0 | 1.5 |
| Road traffic accidents prediction modelling: An. . . [58] | 1.0 | 1.0 | 0.0 | 0.0 | 0.0 | 2.0 |
| Road Traffic Injury Prevention Using DBSCAN. . . [59] | 0.0 | 0.5 | 0.5 | 1.0 | 0.0 | 2.0 |
| SDCAE: Stack Denoising Convolutional. . . [40] | 1.0 | 1.0 | 1.0 | 1.0 | 0.0 | **4.0** |
| Stack ResNet for Short-term Accident Risk. . . [41] | 1.0 | 1.0 | 1.0 | 1.0 | 0.0 | **4.0** |
| Support vector machine in crash prediction at. . . [42] | 1.0 | 1.0 | 0.5 | 0.0 | 0.0 | **2.5** |
| TA-STAN: A Deep Spatial-Temporal. . . [43] | 1.0 | 1.0 | 1.0 | 0.0 | 0.0 | **3.0** |
| Traffic accident prediction based on deep. . . [44] | 1.0 | 1.0 | 1.0 | 1.0 | 0.0 | **4.0** |
| Traffic accident prediction model using. . . [45] | 0.5 | 1.0 | 1.0 | 0.0 | 0.0 | **2.5** |
| Traffic accident prediction using 3-D. . . [60] | 0.0 | 0.5 | 0.5 | 0.0 | 0.0 | 1.0 |
| Utilizing Machine Learning Models to Predict. . . [46] | 1.0 | 1.0 | 1.0 | 1.0 | 1.0 | **5.0** |

**Table A2.** Features list.

| ID | Data Source | Type 1 | Type 2 | Variables |
|---|---|---|---|---|
| PS01 | traffic flow | real | variable | vehicle speed, number of vehicles... |
| | traffic accidents | real | variable | date, time, location, vehicles involved... |
| PS02 | traffic flow | real | variable | vehicle speed, flow, and occupancy |
| | traffic accidents | real | variable | time, location, and collision description |
| PS03 | traffic accidents | real | variable | time and location |
| PS04 | weather conditions | real | variable | clear and adverse |
| | traffic flow | real | variable | vehicle speed, number of vehicles... |
| PS05 | weather conditions | real | variable | sun, cloud, rain, snow, fog, sleet... |
| | traffic flow | real | variable | vehicle speed, number of vehicles... |
| PS06 | weather conditions | simulated | variable | rain, snow, and fog |
| | light conditions | simulated | variable | sun, headlights, and streetlight |
| PS07 | weather conditions | real | variable | type and visibility |
| | traffic accidents | | | Not available |
| | traffic flow | real | variable | vehicle speed, volume, and occupancy |
| PS08 | traffic accidents | real | variable | date, time, location... |
| | traffic flow | real | variable | date, time, number of vehicles... |
| PS09 | weather conditions | real | variable | tweets (snow, sleet, fog...) |
| | traffic accidents | real | variable | time, street name, location... |
| | traffic events | real | variable | tweets (closures, incidents...) |
| PS10 | vehicle data | real | static | type and seat belt |
| | driver's data | real | static | age, gender, and education level |
| | weather conditions | real | variable | visibility |
| | light conditions | | | Not available |
| | traffic accidents | real | variable | time, day of week, severity... |
| | road infrastructure | real | static | geometric characteristics |
| | others | real | static | topographic map... |
| PS11 | weather conditions | real | variable | average temperature, precipitation... |
| | traffic accidents | real | variable | date, time, location, collision type... |
| | taxi trips | real | variable | pick-up timestamp, pick-up location... |
| | traffic flow | real | variable | volume |
| | road infrastructure | real | static | road length, road type, and intersections |
| | population | real | static | Not available |
| | others | real | static | land use |
| PS12 | weather conditions | real | variable | temperature, pressure, humidity... |
| | traffic events | real | variable | collision, broken vehicle, congestion... |
| | road infrastructure | real | static | warning, priority, information... |
| PS13 | weather conditions | real | variable | visibility |
| | traffic accidents | real | variable | number of property damage only... |
| | traffic flow | real | variable | average speed limit... |
| | road infrastructure | real | static | road length, curvature... |

**Table A2.** *Cont.*

| ID | Data Source | Type 1 | Type 2 | Variables |
|---|---|---|---|---|
| PS14 | weather conditions | simulated | variable | street identifier, temperature, snow... |
| | traffic accidents | simulated | variable | time, location... |
| | traffic flow | simulated | variable | Not available |
| | road infrastructure | simulated | static | geometric characteristics |
| | points of interest | simulated | static | location |
| PS15 | traffic accidents | real | variable | timestamp, location... |
| | traffic events | real | variable | timestamp, location, and category |
| | points of interest | real | static | place, category, and location |
| PS16 | weather conditions | real | variable | Not available |
| | traffic accidents | real | variable | date, time, city, state... |
| PS17 | weather conditions | real | variable | precipitation, temperature... |
| | traffic accidents | real | variable | time and location |
| | road infrastructure | real | static | speed limits and volume |
| | others | real | static | satellite images |
| PS18 | traffic accidents | real | variable | identifier, timestamp, location... |
| | traffic flow | real | variable | volume, average speed, and occupancy |
| PS19 | weather conditions | real | variable | Not available |
| | traffic accidents | real | variable | time, day, location, number of dead... |
| | traffic flow | real | variable | time, number of lanes, volume, density... |
| | road infrastructure | real | static | road shape and alignment |
| PS20 | traffic accidents | real | variable | time, location, and security level |
| | pedestrian mobility | real | variable | identifier and location |
| PS21 | weather conditions | undefined | variable | categories |
| | light conditions | undefined | variable | categories |
| | traffic accidents | undefined | variable | time, day of week, and collision month |
| | traffic flow | undefined | variable | type and state of the control device... |
| | road infrastructure | undefined | static | speed limit, road type, pavement type... |
| | traffic events | undefined | variable | type |
| PS22 | traffic accidents | real | static | time, location, vehicles involved... |
| | traffic flow | real | variable | average speed, volume, average... |
| | road infrastructure | real | static | road type, road length, tolls... |
| | weather data | real | variable | visibility and road surface |
| PS23 | weather data | real | variable | precipitation, temperature... |
| | traffic accidents | real | variable | time and location |
| | traffic flow | real | variable | speed limits and annual average daily... |
| | population | real | static | Not available |
| | others | real | static | area size of census blocks |
| PS24 | vehicle data | real | static | type |
| | driver's data | real | static | age, gender, education level... |
| | weather conditions | real | variable | sun, fog, rain, snow, storm, dry, wet... |
| | light conditions | real | variable | day or night |

**Table A2.** *Cont.*

| ID | Data Source | Type 1 | Type 2 | Variables |
|---|---|---|---|---|
| PS24 | traffic accidents | real | variable | time, day of week, and vehicle failure |
| | road infrastructure | real | static | road width, imperfections... |
| | others | real | static | topographic map and digital elevation... |
| PS25 | weather conditions | real | variable | type, wind direction and speed... |
| | traffic accidents | real | variable | time, location, collision type... |
| | traffic flow | real | variable | number of vehicles, occupancy... |
| | road infrastructure | real | static | entrance and exit ramp |
| PS26 | vehicle data | real | static / variable | identifier, time, vehicle speed, and type |
| | traffic accidents | real | variable | date, time, location, and collision type |
| PS27 | traffic accidents | real | variable | tweets and police report |
| PS28 | traffic accidents | real | variable | identifier, time, location... |
| | traffic flow | real | variable | device identifier, timestamp... |
| PS29 | weather conditions | real | variable | precipitation, snowfall, temperature... |
| | road infrastructure | real | static | number of lanes, road type... |
| | pedestrian mobility | real | variable | people's arrivals and departures |
| | points of interest | real | static | place, location, and category |
| | population | real | static | Not available |
| | others | real | static | weekends and holidays |
| PS30 | traffic accidents | real | variable | number of dead and victims... |
| | road infrastructure | real | static | road width, segment length... |
| | population | real | static | Not available |
| | others | real | static | geographical area, income... |
| PS31 | vehicle data | real | variable | location, pick up and pick off time |
| | weather conditions | real | variable | date, time, location, temperature... |
| | traffic accidents | real | variable | time, place, street, collision reason |
| | road infrastructure | real | static | geometric characteristics |
| | taxi trips | real | static | taxi zones |
| | points of interest | real | static | name, location, and category |
| | others | real | variable | start and end point |
| PS32 | weather conditions | real | variable | temperature, dew point, humidity... |
| | traffic accidents | real | variable | time and location |
| | road infrastructure | real | static | name and points for roads... |
| | taxi trips | real | variable | time, location, and speed |
| | points of interest | real | static | name, location, and category |
| PS33 | vehicle data | simulated | variable | speed and vehicle condition |
| | driver's data | simulated | static | age, involvement of alcohol and drugs |
| | weather conditions | simulated | variable | Not available |
| PS34 | traffic accidents | real | variable | age, gender, injury, collision year... |
| | traffic flow | real | variable | volume |
| | road infrastructure | real | static | geometric characteristics (speed limits) |

**Table A3.** Datasets and simulators.

| ID | Datasets/Services | Simulators |
|---|---|---|
| PS01 | Metropolitan Expressway Company Limited and Vehicle Collision and Normal Traffic Condition [61] | |
| PS02 | California Department of Transportation and Highway Performance Measurement System [62] | |
| PS03 | Beijing traffic accident data | |
| PS04 | The Statewide Integrated Traffic Records System [63], Highway Performance Measurement System [64], and Interstate 880 Highway | |
| PS05 | Interstate 15 Highway | |
| PS06 | | Prescan [65] and Matlab/Simulink |
| PS07 | Virginia Department of Transportation [66] and Interstate 64 Highway | |
| PS08 | Intelligent Transportation Systems and Real-Time Monitoring System [67] | |
| PS09 | Twitter API [68] and New York City Open Data [69] | |
| PS10 | National Cartographic Centre [70], Ministry of Roads and Urban Development [71], Meteorological Organization [72], and Highway Police [73] | |
| PS11 | New York of Police Department (Vehicle collisions) [69], New York City Taxi and Limousine Commission, Taxi GPS Data [74], New York City Department of Transportation [75], United States Census Bureau (TIGER files) [76], New York City Department of City Planning [77], and National Climatic Data Center [78] | |
| PS12 | US-Accidents dataset [79], MapQuest Traffic [80], and Microsoft Bing Map Traffic [81] | |
| PS13 | Washington State Department of Transportation [82], Highway Safety Information System [83], and Digital Roadway Interactive Visualization and Evaluation Network [84] | |
| PS14 | | Paramics Microsimulation [85], AIMSUN [86], and VISSIM [87] |
| PS15 | New York Police Department (Traffic Accident Dataset) [69], Points of Interest from New York City [88], and New York City's governmental platform [88] | |
| PS16 | US Accidents (A Countrywide Traffic Accident Dataset) [89] and The Weather Channel [90] | |
| PS17 | Iowa Department of Transportation [91], Iowa Department of Transportation (RWIS) [92], Iowa Department of Transportation (Iowa DOT GIS) [93], and Google Earth Satellite Image [94] | |
| PS18 | Iowa Department of Transportation (Traffic Management Centers Reports) [91], Iowa DOT (Interstate 235 (I-235) and Traffic Flow) [91] | |
| PS19 | Korea Expressway Corporation (Traffic Flow) [95] and Korean National Policy Agency (Traffic Accidents) [96] | |
| PS20 | Japan traffic accident data and Japan human mobility data | |
| PS21 | Not available | |
| PS22 | Signal Four Analytics [97], Central Florida Expressway Authority [98], and National Climatic Data Center (Weather Data) [78] | |

**Table A3.** *Cont.*

| ID | Datasets/Services | Simulators |
|---|---|---|
| PS23 | Iowa Department of Transportation (Vehicle collisions) [91], Stage IV radar rainfall [99], Iowa Department of Transportation (RWIS) [92], Iowa Department of Transportation (Iowa DOT GIS) [93], and Census Data [76] | |
| PS24 | Iran National Cartographic Center [70], Ministry of Roads and Urban Development Islamic Republic of Iran [71], Iran Meteorological Organization [72], National Geographical Organization of Iran [100], and the Information and Technology Department of the Iranian Traffic Police [73] | |
| PS25 | Signal Four Analytics [97], National Climatic Data Center [78] and Central Florida Expressway Authority [98] | |
| PS26 | Autopista Central [101] and Department of Geophysics of University of Chile [102] | |
| PS27 | New York City Police Department (Public Traffic Accident Report) [103] | |
| PS28 | Xiamen traffic accident data and Vehicle License Plate Recognition sensors | |
| PS29 | New York City data | |
| PS30 | Florida Department of Transportation (Crash Analysis Reporting System) [104], FDOT (Roadway Characteristics Inventory) [104], Map of Hillsborough, and United States Census Report [76] | |
| PS31 | New York of Police Department (Vehicle collisions) [69], New York City Taxi and Limousine Commission (Trip Data) [74], National Climatic Data Center (Weather Data) [78], and New York City Open Data [69] | |
| PS32 | Beijing's datasets about traffic accidents and Weather Underground [105] | |
| PS33 | Questionnaires filled by drivers, pedestrians, and others | |
| PS34 | Office of Highway Safety Planning (Michigan Traffic Crash Facts Dataset) [106] | |

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
