# Peer review of "A Systematic Literature Review of Learning-Based Traffic Accident Prediction Models Based on Heterogeneous Sources"

_applsci, doi:10.3390/app12094529_

Round 1

Reviewer 1 Report

The text is clear and easy to read. The main question addressed by the research is relevant and interesting.

The paper is well presented. The presentation of the tables can be improved.

The conclusions are consistent with the evidence and arguments presented.

Reviewer 2 Report

Dear Authors,

I enjoyed reading your paper.

Still, there are some issues to deal with.

For instance:

  • English language and style issues - Grammarly (https://app.grammarly.com) on default settings detected only for the text block resulting from the concatenation of Title+Abstract+Keywords two (2) critical/correctness issues and five (five) more advanced ones (see all of them in the capture at https://tinyurl.com/2p8fdw82). This means a total score of 97 out of a maximum of 100 for this sample above. Still, since you do not appear to be native English speakers, I recommend a revision of the English language and style for the entire article (a complete Grammarly report for the entire paper with the maximum score / the message "no issues found" would be enough);
  • Given the fact that this is a review paper, more references to the existing research are required;
  • The Header contains the collocation “Journal Not Specified”. Of course, this should be replaced;
  • Figure 1 needs a higher resolution (minimum 1000 pixels width/height, or a resolution of 300 dpi or higher);
  • There are so many tables (8) in the paper. Some of them (not essential for understanding the main content) should be moved to the Appendix section. If not existing, this section must be created;
  • Why do the assessment questions AQ01 - AQ04 (pp.4-5) use different threshold values (1 vs. 2)? Is there a scientific basis for these values?
  • Regarding these assessment questions (AQ01 - AQ04), why not include the “missing data treatment” feature/question (e.g. AQ05)? The latter is an important criterion that may lead to major differences in terms of accuracy of classification, the magnitude of influences, etc.
    (Pan et al., 2015, https://doi.org/10.1007/s10489-015-0666-x)
    (Wang et al., 2019, https://doi.org/10.1007/978-3-030-14657-3_7)(Homocianu et al., 2020, https://doi.org/10.3390/app10072573)
    (Idri et al., 2020, https://doi.org/10.1007/s11517-020-02266-x)(Nugroho et al., 2021, https://doi.org/10.1186/s40537-021-00424-y)(Nagarajan & Babu, 2022, https://doi.org/10.1016/j.artmed.2021.102214).

Thank you for your contribution!

Sincerely,

D.

Reviewer 3 Report

This study provides a literature review of learning-based traffic accident prediction models based on heterogeneous sources to learning-based traffic accident predictions.

The article is a description of extensive research on an important subject, and a good research method has been applied. However, at the end of the study, it doesn’t feel like all of the study goals are full filled.

  1. In the introduction, it is said that the article “includes a comparative study of models, selection algorithms, evaluation metrics, and the percentage of data used for training/validation/testing.” In this sense, in most cases the authors provide lists of all the applied models, algorithms, evaluation metrics, etc., without providing a comparative analysis of the considered prediction models, algorithms etc. Shortcomings of prediction models are listed in general. It is said in the Discussion: “Therefore, to compare models to find the one with the best performance is not necessarily real because the results are not completely comparable among the studies.”-perhaps this conclusion should be more elaborated, with some insight into how to overcome this problem.
  2. The relevance of the human factors to the prediction models should be clarified.
  3. Most of the article is written in a good manner however, in parts, it seems like some sentences are incomplete, example line 436/437 “In this context, some solutions for the issues are presented as follows. Thus, the efficient handling of large-scale datasets may be solved using feature extraction”. There are several examples like this in the article.

Minor changes suggestions:

  1. The abstract is too long, some sentences belong to introduction.
  2. Table 5 is not addressed in the article.
  3. Table 1. It is not clear why the total number of articles within one database search engine is calculated as a sum SS01+ SS02+ SS03 given the applied lists of search strings and logical operators. It seems that the search SS03 include results of searches SSO2 and SS01, and SSO2 includes results of SSO1?

Author Response

Please the attachment.

Round 2

Reviewer 3 Report

The authors have satisfactorily responded to all my questions, and made the necessary changes to the manuscript.

This manuscript is a resubmission of an earlier submission. The following is a list of the peer review reports and author responses from that submission.

Round 1

Reviewer 1 Report

In this manuscript, the authors propose a survey on traffic accident prediction with its different solutions. However, this article does not belong to the main topics of the Special Issue “Machine Learning for Cybersecurity Threats, Challenges, and Opportunities II“, and the authors must find a Special Issue according to the main topic of the article. In addition, it is the second time that the authors submit the same article to the same Special Issue. First time under Manuscript ID: applsci-1511729, and they haven't fixed any of the changes suggested by the 3 reviewers rejecting this article. Therefore, this article must be rejected, for those two main drawbacks.

Reviewer 2 Report

In line 113, the authors simply delete the first item of the previous inclusion criteria "IC01. Written in English", which is not the same as a modification or description. Could the author please explain this?

Reviewer 3 Report

The authors propose a systematic literature review about traffic accident prediction models based on heterogeneous sources. The idea sounds interesting and scientific. The main highlighted points follow such as:

1) The abstract is a bit fuzzy, and it does not highlight the proposal's innovation when compared with the same literature papers. This occurs because the authors do not clarifty the gaps under the envisaged problem. 

2) It is not clear in the Introduction Section the innovation of the paper when compared with similar contributions in the literature. 

3) The envisaged problem is not only solved by technological issues. There are diverse human factors that contribute to that solutions. The authors should discuss these aspects in Section 1. 

4) Pages 2 to 18 are focused on presenting the materials and methods used in the paper. Despite being attractive, the information is very lengthy. The authors suggest reducing that section and splitting not essential information for an appendix. Figure 1 and research questions are the most of Section 2. 

5) The essential of the paper is centralized in Sections 3 and 4. The results presented in the section are a bit obvious and only a single summary of what occurred in the previous articles. Only the quantitative results obtained in the evaluation metrics used in the previous paper do not translate the big picture about the envisaged problem. 

6) The depth of discussions presented in Section 4 are limited. Being a systematic review paper, it would expect a deep discussion about the limitation of the area and an indication/guide of future directions. 

7) Overall the idea of the paper is interesting, but most part of the presentation is focused only on the verbosity of the method to filter the paper. The discussion about the results and a deep discussion about the envisaged problem is limited. Additionally, there is no indicated previous paper in the literature proposing the same strategy. Thus, the innovation of the proposal is not clear.